# Impact of Adjuvant Atezolizumab on Recurrences Avoided and Treatment Cost Savings for Patients with Stage II-IIIA Non-Small Cell Lung Cancer in Canada

Quincy Chu [1,*], Kaushik Sripada [2], Sarah Vaselenak [2]🆔, Nick Jovanoski [3] and Melina Arnold [3]🆔

1    Cross Cancer Institute, University of Alberta, Edmonton, AB T6G 1Z2, Canada
2    Hoffmann-La Roche, Mississauga, ON L5N 5M8, Canada; kaushik.sripada@roche.com (K.S.);
    sarah.vaselenak@roche.com (S.V.)
3    F. Hoffmann-La Roche Ltd., 4052 Basel, Switzerland; nick.jovanoski@roche.com (N.J.);
    melina.arnold@roche.com (M.A.)
\*    Correspondence: quincy.chu@albertahealthservices.ca

**Abstract:** This epidemiological model forecasted reductions in recurrences and recurrence treatment cost savings with adjuvant atezolizumab vs best supportive care among Canadians with stage II-IIIA non-small cell lung cancer (NSCLC) at national and provincial levels. The population had resected, programmed cell death 1 ligand 1 (PD-L1)–high (≥50%), *EGFR*−, *ALK*−, stage II-IIIA NSCLC eligible for adjuvant treatment. Patients with recurrence or death and the costs of treating recurrences were estimated for those receiving adjuvant atezolizumab or best supportive care each year (2024–2034). Proportions of patients expected to be event free up to 10 years after treatment initiation were extrapolated with parametric survival analyses. In the base case analysis, 240 fewer recurrences were estimated to occur over 10 years (2024–2034) with adjuvant atezolizumab vs best supportive care across Canada, with 136 (57%) and 104 (43%) fewer locoregional and metastatic recurrences, respectively. Projected costs of treated recurrences were CAD 33.2 million less over 10 years with adjuvant atezolizumab at a national level (adjuvant atezolizumab, CAD 135.8 million; best supportive care, CAD 169.0 million). This model predicts a considerable long-term reduction in recurrences and substantial treatment cost savings with adjuvant atezolizumab vs best supportive care for patients with PD-L1–high early-stage NSCLC in Canada.

**Keywords:** non-small cell lung cancer; atezolizumab; immunotherapy; recurrence; cost; Canada





## 1. Introduction

With an estimated 30,000 new cases and accounting for 24% of cancer-related deaths in 2022, lung cancer is the leading cause of cancer-related death in Canada [1,2]. Despite population-level decreases in smoking prevalence, the absolute number of new lung cancer cases is expected to continue rising due to a growing and aging population [3]. The majority of cases are non-small cell lung cancer (NSCLC), constituting the most prevalent histological type. Patients diagnosed with stage I, II, or resectable stage III disease [4] are considered to have early NSCLC (eNSCLC) [5,6]. Surgical resection is the primary treatment approach for patients with resectable eNSCLC; some patients will also receive neoadjuvant and/or adjuvant chemotherapy. Cisplatin-based chemotherapy has been the historical standard of care for adjuvant treatment of resectable stage IB-IIIA disease post-surgical resection [4,7]. However, recurrence rates have remained relatively high despite adjuvant chemotherapy treatment, with which 5-year recurrence-free survival is approximately 36% [8,9].

In more recent years, immunotherapy and targeted treatments have been shown to reduce the risk of recurrence in patients with eNSCLC [10–12]. The programmed cell death 1 ligand 1 (PD-L1) immunotherapy atezolizumab demonstrated a statistically significant disease-free survival (DFS) benefit compared with best supportive care (BSC) among

patients with resected stage II-IIIA NSCLC who received adjuvant chemotherapy in the randomized, open-label, phase III IMpower010 clinical trial (NCT02486718) [10,11,13]. In this patient population, 39% of patients receiving atezolizumab and 45% of those receiving BSC had DFS events (hazard ratio, 0.79; 95% confidence interval: 0.64 to 0.96; $p$ = 0.020) [11]. Based on the IMpower010 results, atezolizumab was approved by Health Canada in 2022 as a monotherapy for adjuvant treatment following complete resection and no progression after platinum-based adjuvant chemotherapy for adults with stage II-IIIA NSCLC (Union for International Cancer Control/American Joint Committee on Cancer Staging Manual, 7th edition) with tumors with PD-L1 expression on ≥50% of tumor cells (PD-L1 high) [14].

In Canada, treatment access and reimbursement decisions are largely made at the provincial level because the provinces and territories are responsible for the organization, management, and delivery of care for their residents. The national government does, however, set standards of care, perform health technology assessments, and provide funding and other support to the provinces and territories. Understanding the potential of adjuvant atezolizumab to reduce the risk of recurrence in Canadians with eNSCLC can help both provincial health systems and national stakeholders to anticipate the potential magnitude of clinical benefits on a population level. This study forecasts the population-level reduction in recurrences and corresponding treatment cost savings with adjuvant atezolizumab among Canadians with stage II-IIIA NSCLC at the national and provincial levels.

## 2. Materials and Methods

### 2.1. Model Overview

An epidemiological model was developed to estimate the impact of adjuvant atezolizumab on the occurrence of locoregional and metastatic recurrences in Canadian patients with stage II-IIIA (Union for International Cancer Control/American Joint Committee on Cancer Staging Manual, 7th edition) NSCLC and related savings in treatment costs. Atezolizumab was the only immunotherapy approved in Canada for use in the adjuvant setting when this study was conducted. The size of the target patient population was estimated at a provincial level for each year from 2024 to 2034. The number of patients receiving adjuvant atezolizumab and BSC each year was estimated in scenarios in which atezolizumab would or would not be available and funded as an adjuvant treatment option. The probability of a patient experiencing locoregional/metastatic recurrence or dying was estimated based on the number of years of DFS for each treatment scenario (with or without atezolizumab). Finally, the number of patients who experienced locoregional/metastatic recurrence or died was estimated, and drug acquisition and administration costs of treating recurrence were estimated for each treatment scenario.

### 2.2. Patient Population

The size of the target population was calculated as a function of patients with PD-L1–high (≥50%), *EGFR*−, *ALK*−, resected, stage II-IIIA NSCLC who were eligible to receive adjuvant treatment (Figure S1). Patients with *ALK* and *EGFR* mutations were excluded because clinicians are not expected to treat patients with these mutations with immunotherapy due to a lack of benefit in the metastatic setting. Of Canadian patients with lung cancer, 88% were assumed to have NSCLC [15], of whom 29.8% were assumed to have PD-L1–high disease based on Hwang et al. (2021) [16]. Most patients were assumed to have *ALK*− (99.2%; 0.8% *ALK*+) and *EGFR*− (85.8%; 14.2% *EGFR*+) disease [17]. Stage distribution was based on Evans (2018), with 8.0% of all patients with NSCLC diagnosed with stage II and 11.4% diagnosed with stage IIIA [18]. Most tumors in patients with stage II disease (60%) and patients with stage III (26%) disease were assumed to be resectable [18].

### 2.3. Patients Receiving Adjuvant Treatment

The number of patients receiving adjuvant atezolizumab and BSC after chemotherapy was calculated as a function of the number of patients eligible for adjuvant treatment.

Slightly less than half of patients who underwent resection were assumed to receive adjuvant treatment (stage II, 47%; stage IIIA, 41%) [18]. The base case analysis assumed that 50% of eligible patients would receive atezolizumab following adjuvant chemotherapy; scenario analyses were performed to explore variations on this assumption, in which 25% or 75% of eligible patients would receive adjuvant atezolizumab.

### 2.4. Patients Experiencing Recurrence or Dying

The probability of recurrence or death in each treatment arm was based on the number of years that a patient experienced DFS (Figure S2). The proportions of patients expected to be event free up to 10 years after treatment initiation (beyond the 32-month follow-up period of the IMpower010 clinical trial) were extrapolated using the results from parametric survival analyses, assuming that DFS followed several different statistical distributions. Ultimately, a log-logistic distribution was used based on performance metrics (Akaike information criterion and Bayesian information criterion) and the visual inspection of long-term projections of DFS, with the choice deemed appropriate by clinical experts. The extrapolated DFS curves for each treatment arm are provided in Figures S3 and S4. The log-logistic distribution provided a conservative and appropriate estimate of the proportion of patients likely to be disease free after 10 years, which was consistent with cost-effectiveness analyses performed in the US [19], UK [20], and Spain [21]. Scenario analyses were conducted to explore the impact of using exponential, Weibull, log-normal, and gamma distributions. The generalized gamma and Gompertz distributions were tested but resulted in clinically implausible projections of DFS. The number of patients experiencing recurrence or dying in each treatment arm was calculated as a function of the number of patients receiving adjuvant treatment; their time with DFS and number of years since initiating adjuvant treatment were also considered. Proportions of patients who had death as their DFS event in the IMpower010 trial were also calculated (Figure S5).

Since the model uses data from a period in the IMpower010 trial in which recurrences occur more frequently (in the first years following resection and treatment), the proportion of patients who have recurrence at time points beyond the trial (extrapolated DFS) could be overestimated. The base case analysis assumed that recurrences and disease-related deaths would continue up to year 5 based on published evidence and clinical expert consultation [22]. Maeda et al. (2010) showed that the recurrence-free probability at 5 years after primary tumor resection for patients with NSCLC may vary from approximately 65% to 93% and depend on several factors [23]. The authors did not analyze this specifically in patients who received adjuvant chemotherapy. Sonoda et al. (2019) showed that 6.0% and 2.5% of recurrences occurred at 5 to 10 years and >10 years, respectively, in a sample of patients with NSCLC who underwent curative resection and systematic lymph node dissection [24]. While these studies did not separately evaluate the first 2 years after treatment initiation, expert clinicians confirmed that the proportion of patients who may not be at risk of recurrence could start to increase from year 2 [22].

### 2.5. Clinical Pathways

For patients who received adjuvant atezolizumab, the inputs for the patient population, treatment, and outcomes were applied to 2 clinical (disease/treatment) pathways. Patients who received adjuvant atezolizumab and experienced recurrence ≥12 months after adjuvant treatment were considered eligible for immunotherapy rechallenge (Figure 1A). Those with recurrence within 12 months after adjuvant atezolizumab treatment were considered ineligible for immunotherapy rechallenge (Figure 1B). While a 6-month time window was understood to be used in clinical practice, a 12-month window was used in this study even though it was a more conservative assumption, because the model used annual cycles. Patients receiving BSC (adjuvant chemotherapy) were considered eligible for immunotherapy after recurrence, regardless of when recurrence occurred.

The model accounts for the fact that all patients who have a DFS event may not have recurrence but may die. In the population of interest, 7.2% of patients were assumed to

have death as their first DFS event, 51.6% were assumed to have metastatic recurrence, and 41.2% were assumed to have locoregional recurrence based on the IMpower010 clinical trial (patients with stage II-IIIA, PD-L1–high, *ALK−*, *EGFR−* disease pooled across trial arms; clinical cutoff, January 2021) [22]. Among patients with a recurrence, the model also calculated the proportion of patients with locoregional recurrence relative to all recurrences (44.3%) and those with metastatic recurrence relative to all recurrences (55.7%). Additional inputs for recurrence, outcomes, and treatment are provided in Figure 1 and Table S1 [11,25,26].

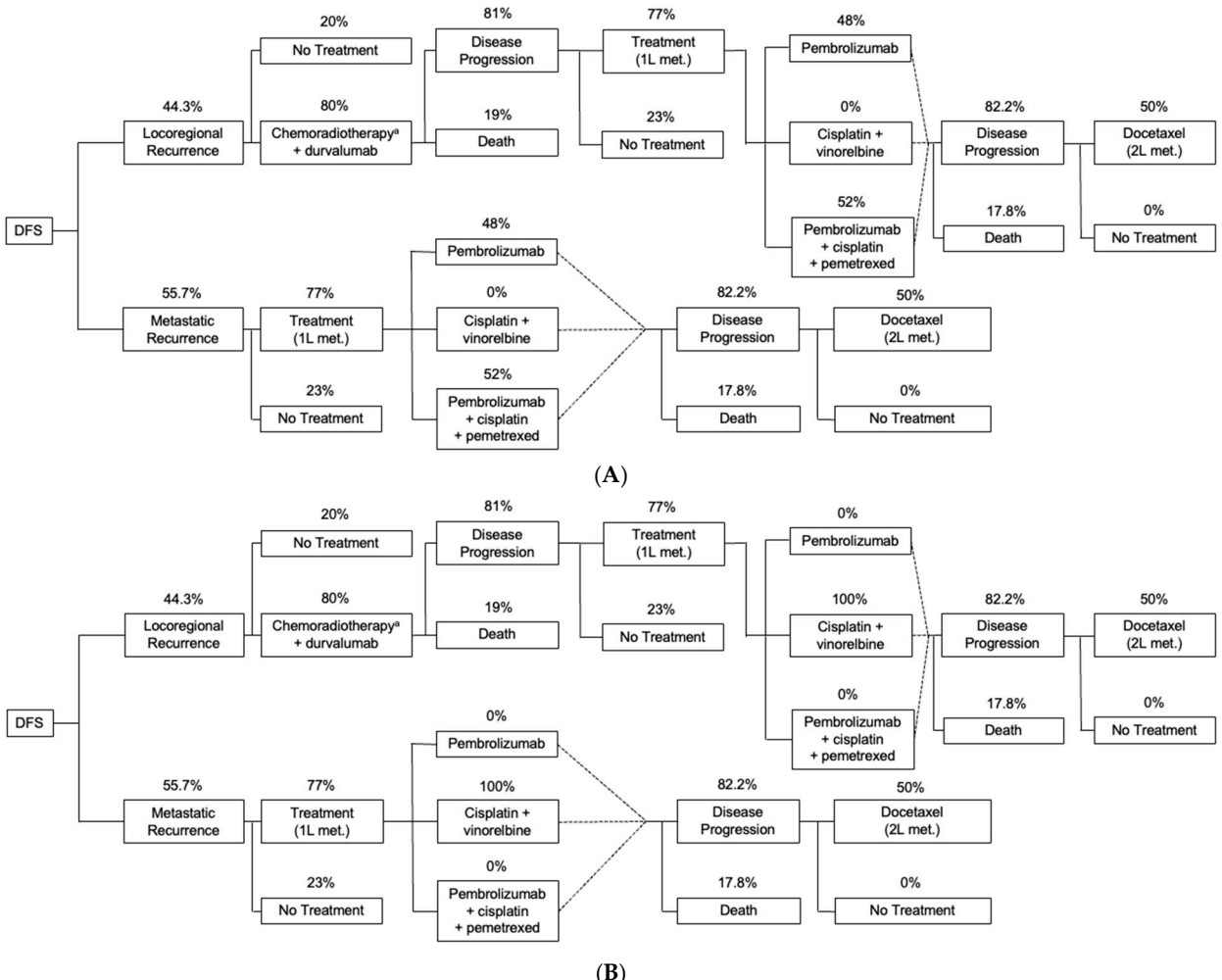

**Figure 1.** Clinical pathways used to inform treatment costs. (**A**) All patients on BSC and patients on adjuvant atezolizumab who experienced recurrence ≥12 months after adjuvant treatment (post recurrence immunotherapy available to both BSC and adjuvant atezolizumab treatment arms). (**B**) Patients on adjuvant atezolizumab who experienced recurrence <12 months after adjuvant treatment (no immunotherapy rechallenge). BSC, best supportive care; DFS, disease-free survival; met, metastatic. [a] The chemotherapy regimen was cisplatin + vinorelbine.

## 2.6. Costs of Treating Recurrence

The model was only designed to estimate the direct costs of treating recurrences (acquisition and administration costs). No other direct costs (e.g., physician services) or indirect costs (e.g., work productivity losses) were included in the calculations. Patients in the atezolizumab or BSC arm who had recurrence ≥12 months after adjuvant treatment and were considered eligible for immunotherapy rechallenge were assumed to have a total cost of treatment for recurrence of CAD 109,340. Patients in the atezolizumab arm who had recurrence within 12 months after adjuvant treatment and were considered ineligible for immunotherapy rechallenge had a total cost of treatment for recurrence of CAD 9257.

The treatment acquisition and administration costs used to derive these cost estimates are provided in Table S2 [27–35]. For costing purposes, patient characteristics, including mean body weight (73 kg) and mean body surface area (1.84 m$^2$), were derived from the IMpower010 trial [11].

### 2.7. Scenario Analyses

Three scenario analyses were performed to explore the implications of varying key model assumptions. The base case analysis assumed that 50% of eligible patients would receive atezolizumab following adjuvant chemotherapy. Scenario analysis 1 explored the impact of 25% or 75% of eligible patients receiving adjuvant atezolizumab. Scenario analysis 2 explored the impact of assuming that recurrences and disease-related deaths would continue up to year 10 rather than year 5. Scenario analysis 3 explored the impact of using the other statistical distributions that were tested for DFS extrapolation (exponential, Weibull, log-normal, and gamma). The generalized gamma and Gompertz distributions were not explored in the scenario analysis because they were clinically implausible.

## 3. Results

### 3.1. Recurrences Avoided with Adjuvant Atezolizumab

Across Canada, the base case analysis estimated 240 fewer recurrences over the 10-year period with adjuvant atezolizumab than with BSC, comprising 136 (57%) fewer locoregional recurrences and 104 (43%) fewer metastatic recurrences as the first event. When evaluated by province, the biggest differences in recurrences over 10 years with adjuvant atezolizumab vs BSC were in Ontario (88 fewer recurrences) and Quebec (67 fewer recurrences) (Figure 2). All estimated recurrences nationally and by province are provided in Table S3. The base case analysis estimated 19 fewer deaths with adjuvant atezolizumab than with BSC, yielding a total of 259 events (recurrences or deaths) avoided over a 10-year period. All estimated deaths and events avoided are provided in Table S3.

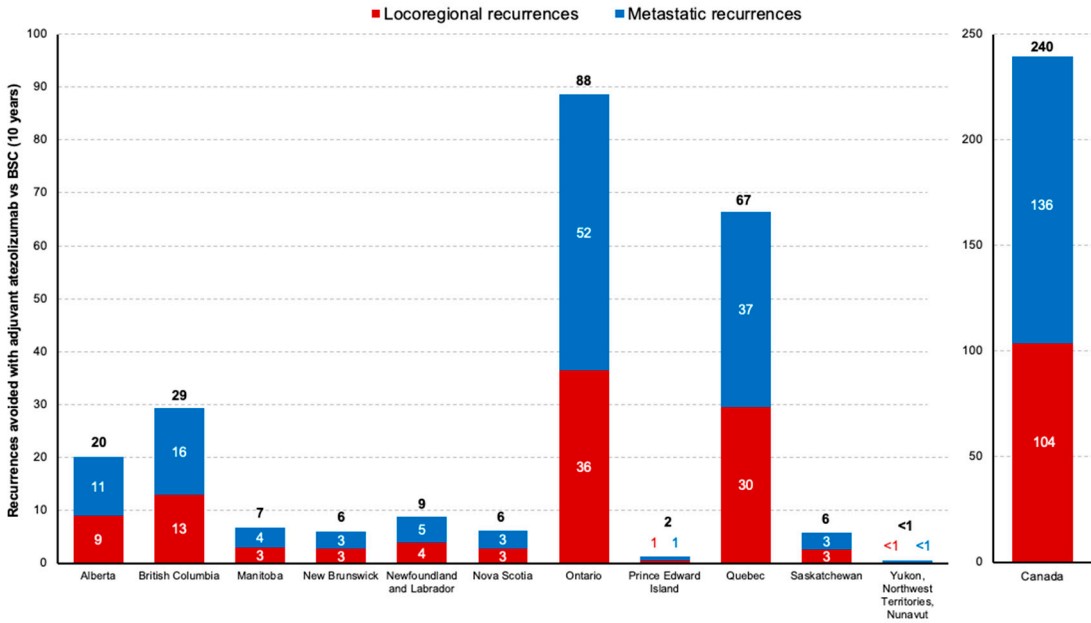

**Figure 2.** Estimated number of avoided recurrences with adjuvant atezolizumab vs BSC by province and recurrence type (10 years, 2024–2034). BSC, best supportive care.

### 3.2. Costs of Treating Recurrences

Based on the estimated differences in recurrences, the base case analysis projected costs of treated recurrences to be CAD 33.2 million less over 10 years with adjuvant atezolizumab at a national level (adjuvant atezolizumab, CAD 135.8 million; BSC, CAD 169.0 million).

As with avoided recurrences, Ontario (CAD 11.7 million) and Quebec (CAD 9.5 million) were estimated to have the greatest cost savings for treated recurrences with adjuvant atezolizumab vs BSC overall (Figure 3). All estimated costs are provided in Table S4.

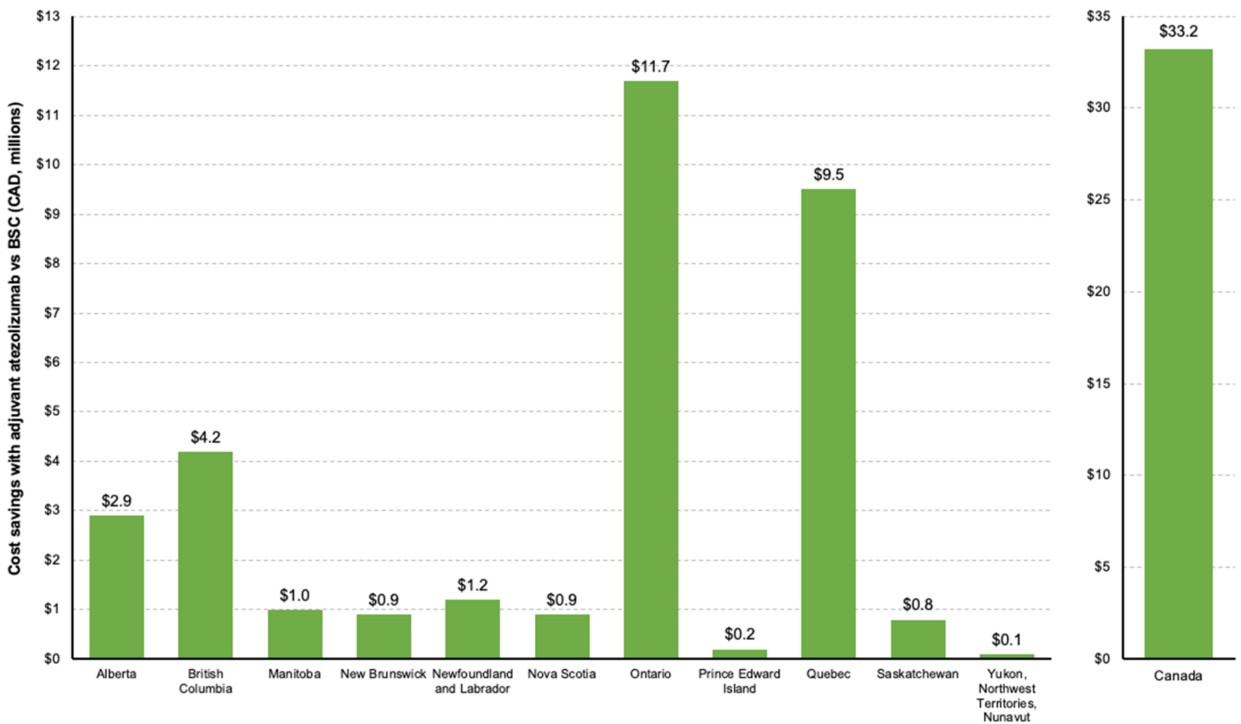

**Figure 3.** Estimated differences in costs of treated recurrences with adjuvant atezolizumab vs BSC by province (10 years, 2024–2034). BSC, best supportive care.

*3.3. Scenario Analyses*

Scenario analysis 1, which explored low (25%) and high (75%) proportions of eligible patients receiving adjuvant atezolizumab, showed correspondingly lower and higher differences in recurrences avoided (Figure S6) and cost savings from treated recurrences (Figure S7) with adjuvant atezolizumab vs BSC over 10 years. Across Canada, there were 117 fewer recurrences with adjuvant atezolizumab vs BSC in the low-uptake scenario (25%) and 350 fewer recurrences in the high-uptake scenario (75%), which were associated with cost savings of CAD 16.6 million and CAD 49.8 million, respectively, over 10 years.

Scenario analysis 2 assumed that recurrences and disease-related deaths would continue up to year 10, as opposed to the year 5 assumption used in the base case analysis. Across Canada, there were 201 fewer recurrences over 10 years (base case, 240) with adjuvant atezolizumab vs BSC, comprising 89 (base case, 104) and 112 (base case, 136) fewer locoregional and metastatic recurrences with atezolizumab vs BSC, respectively (Figure S8). Cost savings associated with all avoided recurrences were CAD 29.6 million (base case, CAD 33.2 million) over 10 years (Figure S9).

Scenario analysis 3, which explored avoided recurrences (Figure S10) and cost savings (Figure S11) with adjuvant atezolizumab vs. BSC, supported the methodological decision that the log-logistic distribution was likely to provide the most conservative and clinically reasonable extrapolation of DFS over 5 years.

**4. Discussion**

This study estimated a meaningful impact on recurrences and associated treatment costs for a scenario in which adjuvant atezolizumab is available and funded for patients with eNSCLC in Canada. The base case analysis assumed a 50% uptake of adjuvant atezolizumab starting in 2024 and estimated that 240 recurrences would be avoided among

PD-L1–high patients with stage II-IIIA NSCLC across Canada over 10 years. This corresponded to an 18% decrease in the estimated number of recurrences compared with BSC-only treatment. Nearly half of the estimated avoided recurrences were in the locoregional setting (43%); however, it should be noted that subsequent recurrences were not considered (such as those associated with metastatic progression). Based on the epidemiological inputs and assumptions, the most avoided recurrences were in Ontario and Quebec, followed by British Columbia and Alberta. The variation in the estimated number of avoided recurrences and associated costs across provinces and territories is reflective of differences in disease burden and population size. The estimated costs of treated recurrences were CAD 33.2 million lower when adjuvant atezolizumab was given to 50% of eligible patients. The model only considered recurrence-related treatment acquisition and administration costs, and the inclusion of additional direct and indirect costs may have resulted in more cost savings from avoided recurrences with adjuvant atezolizumab vs BSC. Scenario analyses exploring the impact of a lower (25%) or higher (75%) uptake of adjuvant atezolizumab suggested that 117 to 350 total recurrences would be avoided with adjuvant atezolizumab vs BSC across Canada over 10 years, resulting in a cost savings of CAD 16.6 million and CAD 49.8 million, respectively. Overall, the base case model demonstrated that if only half of qualifying patients receive adjuvant atezolizumab, a substantial number of recurrences would be avoided, potentially leading to substantial cost savings.

Although the cancer immunotherapy treatment landscape continues to evolve, atezolizumab was the only immunotherapy approved in Canada for use in the adjuvant setting when this model was developed. To our knowledge, this is the first estimation of the potential impact of adjuvant immunotherapy treatment on the number of recurrences in Canadian adults with eNSCLC. The clinical impact and cost-effectiveness of avoided recurrences with adjuvant atezolizumab for patients with eNSCLC have been explored in other countries [19,20,36]. In the US, Lee (2022) estimated per-patient per-month costs to be approximately six times (USD 6319) higher for those with resected NSCLC and recurrence than for matched peers who did not have a recurrence [19]. Sharma (2022) further estimated that patients with eNSCLC in the US could avoid 1030 recurrences and 369 deaths with adjuvant atezolizumab treatment, equating to a reduction of USD 785 million in recurrence-related costs over 5 years [36]. While our study was not an all-encompassing cost-effectiveness analysis, the estimated impact of adjuvant atezolizumab in Canada is consistent with analyses in other countries and healthcare settings that have projected meaningful reductions in the burden of recurrences among patients with eNSCLC.

Certain strengths and limitations of the model should be considered when interpreting these findings. This model focused specifically on the Canadian perspective from provincial and national levels and used inputs from the IMpower010 clinical trial, public sources, and the published literature, which were tailored to the Canadian perspective as much as possible. It should be noted that in Canada, patients with recurrences are generally considered to be ineligible for immunotherapy rechallenge unless the recurrence occurs after 6 months. This model used a 12-month cutoff; however, this would not have impacted the estimated number of recurrences. This study assumed no changes in stage distribution and other clinical or epidemiological inputs over time and used the same inputs across provinces, which may be variable. The availability of immunotherapy options such as atezolizumab may lead to increased surgical and adjuvant treatment rates; thus, ignoring temporal changes in these assumptions could have provided a further conservative estimate of the impact of adjuvant atezolizumab in this setting. The authors note that extrapolation of DFS data comes with inherent uncertainty around clinical benefits beyond the timeframe of the IMpower010 trial. However, the log-logistic distribution was a conservative, appropriate approach based on clinical consultation, and scenario analyses illustrate the results based on other possible distributions.

It should be noted that the objective of this model was to evaluate the direct treatment costs associated with treating recurrences and was not designed or intended to account for the costs of adjuvant treatment or the broader societal burden and costs related to

recurrences, such as health utilities and indirect costs. Future work may explore the societal perspective, which may be most relevant at the provincial level. However, expanding the quantifiable burden of recurrences would likely improve the economic benefit of avoided recurrences and, in turn, the health economic value of adjuvant atezolizumab in this setting.

## 5. Conclusions

This study suggested that a 25% to 75% uptake of adjuvant atezolizumab could avoid 117 to 350 total recurrences compared with BSC across Canada over 10 years, resulting in cost savings of CAD 16.6 million to CAD 49.8 million. This model predicts that adjuvant atezolizumab following chemotherapy provides a considerable population-level reduction in recurrences among patients with PD-L1–high eNSCLC and reduces the treatment costs and economic burden of recurrent NSCLC in Canada.

**Supplementary Materials:** The following supporting information can be downloaded at: https://www.mdpi.com/article/10.3390/curroncol31060251/s1, Figure S1. Calculation of the target population. Figure S2. Probability of a patient experiencing recurrence or dying. Figure S3. Atezolizumab disease-free survival extrapolations. Figure S4. Best supportive care disease-free survival extrapolations. Figure S5. Number of patients experiencing recurrence or dying. Figure S6. Estimated number of avoided recurrences with adjuvant atezolizumab compared with BSC by province, recurrence type, and uptake scenario (10 years, 2024–2034). Figure S7. Estimated differences in costs of treated recurrences with adjuvant atezolizumab compared with BSC by province and uptake scenario (10 years, 2024–2034). Figure S8. All recurrences avoided with adjuvant atezolizumab vs BSC where all events continue up to year 10, by province and recurrence type (10 years, 2024–2034). Figure S9. Cost savings with adjuvant atezolizumab vs BSC where all events continue up to year 10, by province (10 years, 2024–2034). Figure S10. All recurrences avoided with adjuvant atezolizumab vs BSC by DFS distribution and province (10 years, 2024–2034). Figure S11. Cost savings with adjuvant atezolizumab vs BSC by DFS distribution and province (10 years, 2024–2034). Table S1. Clinical pathway inputs. Table S2. Locoregional and metastatic recurrence treatment costs. Table S3. Estimated events in the base case analysis (10 years, 2024–2034). Table S4. Estimated costs ($MM) of treating recurrences in the base case analysis (10 years, 2024–2034). References [11,25–35] is cited in the Supplementary Materials.

**Author Contributions:** Q.C.: Conceptualization, investigation, methodology, supervision, writing–review and editing. K.S.: Conceptualization, data curation, formal analysis, investigation, methodology, project administration, resources, supervision, validation, writing—review and editing. S.V.: Conceptualization, data curation, formal analysis, investigation, methodology, project administration, resources, supervision, writing—review and editing. N.J.: Conceptualization, data curation, formal analysis, investigation, methodology, project administration, resources, software, supervision, validation, writing–review and editing. M.A.: Conceptualization, data curation, formal analysis, investigation, methodology, project administration, resources, supervision, writing–review and editing. All authors have read and agreed to the published version of the manuscript.

**Funding:** This study was sponsored by F. Hoffmann-La Roche Ltd.

**Institutional Review Board Statement:** There was no patient-level data collected, generated, or analyzed.

**Informed Consent Statement:** There was no patient-level data collected, generated, or analyzed.

**Data Availability Statement:** No new data were created for this study. This was a calculator based on existing aggregated data from publications and public sources.

**Acknowledgments:** Support for third-party writing assistance for this manuscript, furnished by Jeff Frimpter, MPH, of Nucleus Global, an Inizio Company, was provided by F. Hoffmann-La Roche Ltd.

**Conflicts of Interest:** Employees of the study sponsor are co-authors of this paper who participated in the choice of the study; design of the study; collection, analysis, and interpretation of data; writing the report; critical review of the manuscript; and the decision to publish the results. Quincy Chu: Research funding from F. Hoffmann-La Roche Ltd. Kaushik Sripada: Employee of F. Hoffmann-La Roche Ltd. Sarah Vaselenak: Employee of F. Hoffmann-La Roche Ltd. Nick Jovanoski: Employee of F. Hoffmann-La Roche Ltd. Melina Arnold: Employee of F. Hoffmann-La Roche Ltd.

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
