# Peer review of "Impact of Adjuvant Atezolizumab on Recurrences Avoided and Treatment Cost Savings for Patients with Stage II-IIIA Non-Small Cell Lung Cancer in Canada"

_curroncol, doi:10.3390/curroncol31060251_

Round 1
Reviewer 1 Report
Comments and Suggestions for Authors
I have read and reviewed the submitted manuscript " Impact of adjuvant atezolizumab on recurrences avoided and treatment cost savings for patients with stage II-IIIA non-small cell lung cancer in Canada”. I’d like to commend the authors for this well written paper. They have explored a part of treatment that until recently gets very little attention, the long-term costs to a health care system when adding drugs that are quite expensive.
Points to consider:
1. Many readers may not fully understand how health care is delivered in Canada. If possible, a very short description may be of benefit in the Discussion. This helps to put into context the results and how translatable it may be for readers from other countries and the possible impacts on their own health system.
2. The authors may want to consider adding a sentence to the CONCLUSION section that provides the range for cost savings over 10 years between 25% and 75% uptake of adjuvant atezolizumab to reinforce their findings.
Author Response
- Many readers may not fully understand how health care is delivered in Canada. If possible, a very short description may be of benefit in the Discussion. This helps to put into context the results and how translatable it may be for readers from other countries and the possible impacts on their own health system.
Authors’ response: We appreciate this point and have added the following content to the Introduction (p4, ln 65-71): “In Canada, treatment access and reimbursement decisions are largely made at the provincial level because the provinces and territories are responsible for the organization, management, and delivery of care for their residents. The national government does, however, set standards of care, perform health technology assessments, and provide funding and other support to the provinces and territories. Understanding the potential of adjuvant atezolizumab to reduce the risk of recurrence in Canadians with eNSCLC can help both provincial health systems and national stakeholders to anticipate the potential magnitude of clinical benefits on a population level. This study forecasts…” - The authors may want to consider adding a sentence to the CONCLUSION section that provides the range for cost savings over 10 years between 25% and 75% uptake of adjuvant atezolizumab to reinforce their findings.
Authors’ response: We have added a brief summary statement to the Conclusion as suggested by the reviewer (p16, ln 317-319): “This study suggested that a 25% to 75% uptake of adjuvant atezolizumab could avoid 117 to 350 total recurrences compared with BSC across Canada over 10 years, resulting in a cost savings of $16.6 million to $49.8 million.”
Reviewer 2 Report
Comments and Suggestions for Authors
The clinical impact and cost-effectiveness of avoided recurrences with adjuvant atezolizumab for patients with early-stage NSCLC have been explored in US and UK. This study firstly analyzed whether adjuvant atezolizumab reduces recurrence and is cost-effective in Canada.
Overall, this paper is well written and there is not much that needs to be modified.
Figure 2, Line 236-238
The authors should provide additional explanation as to why the estimated number of avoided recurrences due to atezolizumab varies greatly across Canadian provinces.
There are few figures and tables in the paper. The authors provided many supplementary figures and tables, but important figures and tables should be included in the main text.
Author Response
- Figure 2, Line 236-238. The authors should provide additional explanation as to why the estimated number of avoided recurrences due to atezolizumab varies greatly across Canadian provinces.
Authors’ response: To the reviewer’s point, we have added the following content to the Discussion (p13, ln 259-260): “The variation in the estimated number of avoided recurrences and associated costs across provinces and territories is reflective of differences in disease burden and population size” - There are few figures and tables in the paper. The authors provided many supplementary figures and tables, but important figures and tables should be included in the main text.
Authors’ response: We appreciate the reviewer’s point, however, we would like to keep the supplementary figures in the supplement as they are not part of the base case analysis and may confuse readers who are less well versed with these types of calculators. The figures in the supplement are the extrapolated disease-free survival curves from all explored distributions (including those that were not used) and variations on the base case assumptions generated for the scenario analyses. While we understand the reviewer’s aim is to provide more information in the body of the paper, we would ideally prefer to keep the main body focused on the base case findings.